# De novo diploid genome assembly using long noisy reads

Fan Nie[1,2,3,8], Peng Ni [1,2,4,8], Neng Huang[1,2,4], Jun Zhang[1,2,4], Zhenyu Wang[5], Chuanle Xiao [6] ✉, Feng Luo [7] ✉ & Jianxin Wang [1,2,4] ✉

The high sequencing error rate has impeded the application of long noisy reads for diploid genome assembly. Most existing assemblers failed to generate high-quality phased assemblies using long noisy reads. Here, we present PECAT, a **P**hased **E**rror **C**orrection and **A**ssembly **T**ool, for reconstructing diploid genomes from long noisy reads. We design a haplotype-aware error correction method that can retain heterozygote alleles while correcting sequencing errors. We combine a corrected read SNP caller and a raw read SNP caller to further improve the identification of inconsistent overlaps in the string graph. We use a grouping method to assign reads to different haplotype groups. PECAT efficiently assembles diploid genomes using Nanopore R9, PacBio CLR or Nanopore R10 reads only. PECAT generates more contiguous haplotype-specific contigs compared to other assemblers. Especially, PECAT achieves nearly haplotype-resolved assembly on *B. taurus* (Bison×Simmental) using Nanopore R9 reads and phase block NG50 with 59.4/58.0 Mb for HG002 using Nanopore R10 reads.

De novo genome assembly is a fundamental task in genomic research[1–3]. Using long noisy reads from sequencing technologies, such as complete long reads (CLR) sequencing of Pacific Biosciences (PacBio) and Nanopore sequencing of Oxford Technologies (ONT), many assemblers[4–9] now can effectively reconstruct high-quality genome sequences for haploid or inbred species. However, a significant fraction of genetic information of a diploid genome is lost in those assemblies and most of them have lots of haplotype switch errors. Because long noisy reads, such as PacBio CLR reads and Nanopore reads, usually contain a 5–15% sequencing error[10], it is difficult to distinguish heterozygotes from sequencing errors in long noisy reads[11–15], which prevents the diploid assemblers from generating long haplotype-specific contigs. Recently, by combining long noisy reads with additional highly accurate sequencing data, such as parental short reads[16–18], PacBio HiFi reads[17,18], Hi-C reads[19], Strand-seq data[20], or gamete cell data[21], assemblers now can produce more contiguous haplotype-specific contigs. However, the requirements for additional sequencing data increase costs and limit their applications in practice[22,23]. Meanwhile, it is easier to identify haplotype differences using high accurate PacBio HiFi reads[24] (< 1% error rate), which has been widely used for haplotype resolved assemblies[25–28]. However, the average lengths of HiFi reads (10–25 kb) are shorter than those of long noisy reads. For example, the ultra-long reads of Nanopore are up to 1 M in length and with read N50 > 100 kb[29]. Longer reads usually help to assemble more contiguous contigs[30]. Therefore, it will be useful to develop an assembler that can take advantage of long noisy reads to generate more contiguous haplotype-specific contigs for diploid genomes.

To achieve high-quality assemblies, error correction is usually a useful step for genome assembling using long noisy reads. One

[1]School of Computer Science and Engineering, Central South University, Changsha 410083, China. [2]Xiangjiang Laboratory, Changsha 410205, China. [3]National Center for Applied Mathematics in Hunan and Key Laboratory of Intelligent Computing and Information Processing of Ministry of Education, Xiangtan University, Xiangtan, Hunan 411105, China. [4]Hunan Provincial Key Lab on Bioinformatics, Central South University, Changsha 410083, China. [5]Institute of Nanfan & Seed Industry, Guangdong Academy of Sciences, Guangdong 510316, China. [6]State Key Laboratory of Ophthalmology, Zhongshan Ophthalmic Center, Sun Yat-sen University #7 Jinsui Road, Tianhe District, Guangzhou, China. [7]School of Computing, Clemson University, Clemson, SC 29634-0974, USA. [8]These authors contributed equally: Fan Nie, Peng Ni ✉e-mail: xiaochuanle@126.com; luofeng@clemson.edu; jxwang@mail.csu.edu.cn

challenge for correct-then-assemble pipelines in assembling a diploid genome using long noisy reads is how to retain heterozygotes during error correction. If the sequencing error rate exceeds haplotype divergence, current correct-then-assemble pipelines[4–6,9] eliminate heterozygotes as sequencing errors or mixed alleles of different haplotypes in a read. Therefore, corrected reads don't contain heterozygote information for haplotype phasing. Assemblers, such as FALCON-Unzip[4], used raw reads instead of corrected reads in their "unzip" phasing step, then mapped the raw reads to the corrected reads for the later diploid assembly. This has led to assembly errors due to the heterozygote errors in the corrected reads and mapping errors between corrected reads and raw reads as the length of reads and their SNP position changed after correction[4]. Furthermore, phasing raw reads is more difficult due to their high error rate.

Another challenge is partitioning reads according to their haplotypes in the phasing step. After error correction, there still are sequencing errors left in corrected reads. The error profile of corrected reads is also more complex than that of PacBio HiFi reads. Due to the sequencing errors, a significant fraction of reads is difficult to be phased to the correct haplotypes. This leads to inconsistent overlaps[25] among reads, whose reads come from different haplotypes or different copies of the segmental duplication. The inconsistent overlaps can cause haplotype switch errors or unresolved repeats. Accurately phasing the reads, and then identifying and removing inconsistent overlaps is important in overlap-graph-based diploid genome assembly. Even if we assemble the genome using PacBio HiFi reads, there is still need error correction step to improve the phasing accuracy[25]. The higher error rate in corrected long noisy reads makes phasing and identifying inconsistent overlaps more difficult. It is necessary to design a more accurate and robust method to identify inconsistent overlaps for long noisy read-based assemblers.

Furthermore, to construct two haplotypes of diploid genomes, the assemblers often need higher coverage of sequencing data. However, with the increasing amount of sequencing data, the running time of assembly increases nonlinearly. Therefore, assembling large diploid genomes at an acceptable time is another challenge. Overlap-graph-based assemblers have their unique advantages to assemble diploid genomes since the overlaps can be reused in subsequent steps once they are found. This helps to design an efficient multi-round assembly strategy. The overlap-graph-based assemblers usually use seed (i.e., k-mers, minimizers[31]) based methods to find candidate overlaps, then perform local alignment to find the true overlaps. The local alignment is the major computational bottleneck of overlap-graph-based approaches. Although skipping local alignment can accelerate the overlap-finding step[32], it leads to lots of false-positive overlaps and introduces errors in assembly graphs. String-graph-based[33] approaches, a type of overlap-graph-based approach, ignore the reads contained by other reads and mark most edges in the graph as transitive edges[33] that don't contribute to the construction of contigs. Then, it is not necessary to perform local alignment on those read pairs. Therefore, it is possible to design a fast string-graph-based diploid genome assembler by minimizing the number of local alignments needed.

In this work, we present PECAT, a **P**hased **E**rror **C**orrection and **A**ssembly **T**ool, designed to reconstruct diploid genomes from long noisy reads, including PacBio CLR reads and Nanopore reads. PECAT follows the correct-then-assemble strategy, including a haplotype-aware error correction module, which can retain heterozygote alleles while correcting sequencing errors, and a two-round string graph-based assembly module. To accelerate the assembling, PECAT only performs local alignment when it is necessary instead of performing local alignments on all candidate overlaps. PECAT outputs two sets of contigs either in primary/alternate format (long contigs with the mosaic of homologous haplotypes and short haplotype-specific contigs) or in the dual assembly format[27] (two sets of long contigs with the mosaic of homologous haplotypes). PECAT can efficiently assemble

diploid genomes using Nanopore R9, PacBio CLR or more accurate Nanopore R10 reads only and generate more contiguous haplotype-specific contigs compared to other assemblers.

## Result

### Haplotype-aware error correction

Our error correction method (Fig. 1a and Supplementary Fig. 1) is based on the partial-order alignment (POA) graph[34] method. For the template read to be corrected, a POA graph is built from the alignment of supporting reads. Then, the path with the highest weight is found in the POA graph to construct the consensus sequence for the template read. However, if the sequencing error rate exceeds haplotype divergence, current methods inevitably select supporting reads from different haplotypes. This causes either heterozygote alleles in the template reads to be eliminated as sequencing errors or heterozygous alleles from different haplotypes to be mixed in corrected reads.

To maintain the heterozygote alleles in the template read, we need to select supporting reads from the same haplotype to correct it. After error correction, there still are sequencing errors left in corrected reads. After analyzing the POA graph, we have found that the difference between random sequencing errors and heterozygotes can be reflected in the POA graph. In the case of heterozygotes, there are two dominant parallel branches in the POA graph. In the case of random errors, there tends to be only one dominant branch (Supplementary Fig. 1b). Based on this finding, we design a scoring algorithm to estimate the likelihood that the supporting read and the template read are from the same haplotype (**Methods**). For each position at which there are two dominant parallel branches, if the supporting read and the template read pass through the same dominant branch, the score is increased by 1 and if the supporting read and the template read pass through the different dominant branches, the score is decreased by 1 (Supplementary Fig. 1c). A higher score means that the template read and the supporting read are more likely from the same haplotype. If the supporting reads come from different haplotypes, the histogram of scores of all supporting reads should show two peaks (Supplementary Fig. 1d). Then, we select the high-scoring supporting reads, which are very likely to be in the same haplotype as the template read, to correct the template read. To further increase the likelihood of selecting supporting reads in the same haplotype, we assign different weights to the reads according to their score. The higher the score of a read, the larger its weight is assigned (**Methods**). Then, we remove unselected reads in the POA graph by assigning their weights to 0. Finally, we use a dynamic programming approach to find the path with the highest weight in the POA graph and concatenated the nodes in the path, and generate the consensus sequence (**Methods**).

We evaluate the performance of the selection method on seven diploid datasets of *S. cerevisiae* (SK1×Y12), *A. thaliana* (Col-0×Cvi-0), *D. melanogaster* (ISO1×A4), *B. taurus* (Angus×Brahman), *A. thaliana* (Col-0×C24), *B. taurus* (Bison×Simmental) and HG002 (Nanopore R9), all with reads classified by the trio-binning algorithm[16] (**Methods**). As shown in Supplementary Table 1, without the selection method, 36.7%, 31.5%, 41.1%, 28.1%, 33.8%, 37.2% and 27.4% of supporting reads are inconsistent reads, respectively, which are from the haplotype different from the one of template read. Meanwhile, there are only 2.8%, 3.5% 4.9%, 3.6%, 4.3%, 2.9%, and 4.2% inconsistent reads in selected supporting reads, respectively. After further re-weighting the scores, the percentages of inconsistent reads are reduced to 2.1%, 3.1%, 4.0%, 3.1%, 3.5%, 2.3%, and 3.4%, respectively.

Since most of the selected supporting reads are from the same haplotype of the template read, most heterozygote alleles in the template read are correctly retained in our error correction method. We compare our method with other methods including Canu[5], MECAT2[6], and NECAT[9] on simulated PacBio CLR data and Nanopore data (Fig. 2a, Supplementary Table 2, and Supplementary Note 1). The overall accuracies of corrected reads are similar, which exceeds 99%. However, the accuracies of SNP alleles in corrected reads are less than

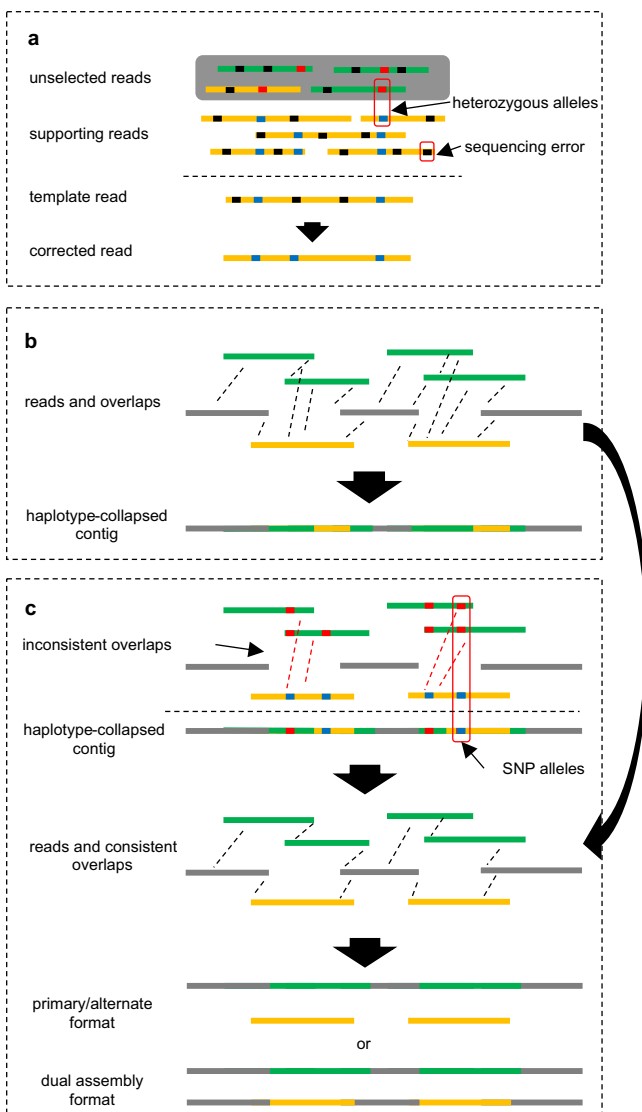

**Fig. 1 | Overview of PECAT. a** Haplotype-aware error correction method. The reads in different colors (in green or in yellow) are from different haplotypes. The reads with a dark gray background color indicate that they are not selected. The supporting reads that are more likely from the same haplotype are selected to correct the template read. **b** The first round of assembly. PECAT finds overlaps between reads. The dashed lines indicate there are overlaps between reads. Our string-graph-based assembler is performed to construct haplotype-collapsed contigs. The reads in green or in yellow are from heterozygosity regions. The same color indicates that the reads come from the same haplotype. The reads in grey are from homozygosity regions. **c** The second round of assembly. PECAT identifies inconsistent overlaps by calling and analyzing SNP alleles in reads. After removing inconsistent overlaps from the overlaps found in the first round of assembly, PECAT performs our string-graph-based assembler again to construct the contigs in the primary/alternate format or the dual assembly format.

80% for Canu and less than 60% for MECAT and NECAT, which are far worse than those of raw reads and reads corrected by PECAT. When the heterozygosity rate of simulated reads is greater than or equal to 0.0005, the accuracy of SNP alleles in reads corrected by PECAT is greater than 99%, which exceeds that of raw reads (96% or 98%). However, when the heterozygosity rate of simulated reads is equal to 0.0001, the accuracy of SNP alleles in reads corrected by PECAT is reduced to 92% ~ 94%, which is less than that of raw reads (96% ~ 98%). Therefore, for high heterozygosity genome regions (>= 0.0005), our haplotype-aware error correction can preserve the SNPs well.

However, for low heterozygosity genome regions (< 0.0005), we need to combine with other methods to improve the accuracy of SNP calling (**Methods**).

## Fast string graph-based assembler

After error correction, PECAT implements two rounds of string graph-based assembly. In each round of assembly, we first construct the overlaps between the corrected reads using the seed-based alignment method (minimap2[35]), which allows us to build the overlaps quickly. However, seed-based alignment can bring low-quality overlaps with low identity or with long overhangs. Those low-quality overlaps could introduce errors during assembling. Performing local alignment on overlaps to identify low-quality ones becomes the major computational cost. To speed up the assembling, PECAT only performs local alignments when it is necessary during the construction of the string graph. (**Methods**). First, to reduce overhangs of overlaps, we use diff algorithm[36] to extend the candidate overlaps to the ends of the reads. Here, we only perform local alignment on overhangs instead of on the whole overlap. We remove the overlaps if their overhangs are still long $(\min(100, 0.01 \cdot l)$ for PacBio reads and $\min(300, 0.03 \cdot l)$ for Nanopore reads, where $l$ is length of the read). On the diploid datasets of *S. cerevisiae* (SK1×Y12), *A. thaliana* (Col-0×Cvi-0), *D. melanogaster* (ISO1×A4), *B. taurus* (Angus×Brahman), *A. thaliana* (Col-0×C24), *B. taurus* (Bison × Simmental) and HG002, 5.3%, 10.3%, 82.2%, 12.8%, 20.0%, 59.4% and 56.3% of candidate overlaps with long overhangs have been considered as low quality and removed, respectively (Supplementary Table 3). Then, we further filter out the overlaps whose reads are contained in other reads or with low coverage. Only 0.15%, 0.18%, 0.86%, 0.86%, 0.03%, 1.23%, and 0.15% of overlaps remained on the above seven diploid datasets (Supplementary Table 3). We then construct a directed string graph from the remaining overlaps. We find the transitive edges using Myers' algorithm[33] and mark them as inactive edges, which are not used for constructing contigs. On the above diploid datasets, only 25.1%, 16.7%, 16.3%, 18.5%, 20.7%, 15.4%, and 23.6% of edges are active (Supplementary Table 3).

To remove low-quality edges in the string graph, we calculate the identity of the overlaps (the active edges) in the string graph using local alignments. Since most edges have been marked as inactive, we only need to perform local alignments for a small portion of overlaps. On the above diploid datasets, 2.1%, 12.2%, 9.5%, 2.2%, 13.2%, 1.8%, and 6.9% of active edges have been removed because their identities are less than the threshold (**Methods**) (Supplementary Table 3). After the above step, some paths in the graphs connected by low-quality edges are broken. Those broken paths need to be connected using the transitive edges that have been labeled as inactive edges. We then select transitive edges with the longest alignment and their identity greater than the threshold to connect the broken paths. On the above diploid datasets, about 0.67%, 0.12%, 0.15%, 0.09%, 0.13%, 0.05%, and 0.16% of transitive edges have been reactivated (Supplementary Table 3). Finally, we attempt to use the contained reads to connect the broken paths, and about 0.30%, 0.39%, 0.17%, 0.11%, 0.77%, 0.13%, and 0.45% of new edges are added to the graph on the above diploid datasets (Supplementary Table 3). In the first round of assembly (Fig. 1b), PECAT finds linear paths from this string graph and constructs haplotype-collapsed contigs. In the second round of assembly (Fig. 1c), PECAT identifies and removes inconsistent overlaps (next Section) in the string graph, and then generates two sets of contigs in primary/alternate format or dual assembly format. After generating contigs, we use corrected reads (CLR data) or raw reads (Nanopore data) to polish them to improve the quality (**Methods**).

## Identification of inconsistent overlaps

The inconsistent overlaps connect reads from different haplotypes or different copies of the segmental duplication in the overlap graph. They cause haplotype switch errors or unresolved repeats

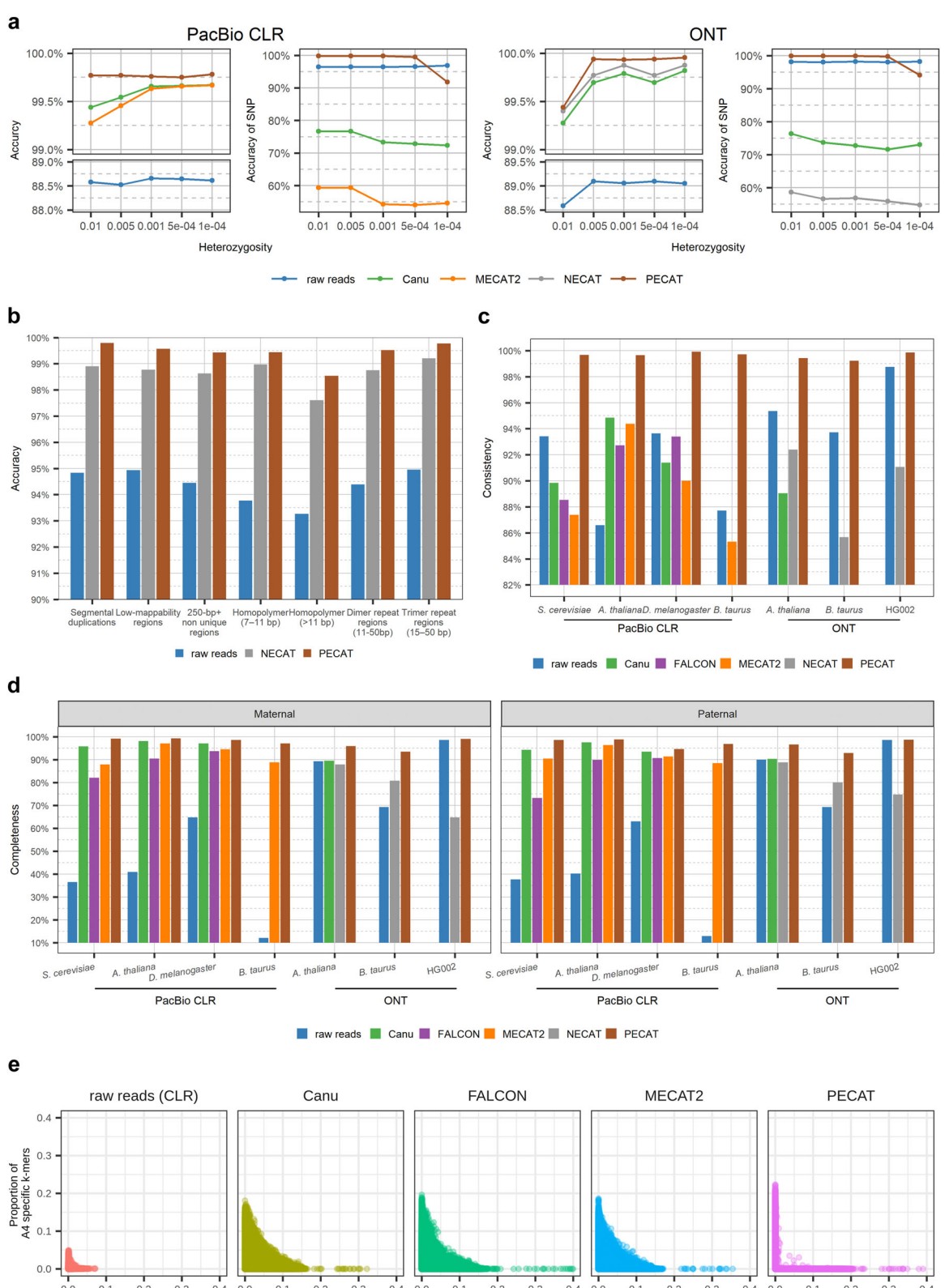

(Supplementary Fig. 2). After identifying and filtering out inconsistent overlaps, only the overlaps between reads from the same haplotype or the same copies of the segmental duplication are left, and then the assembler can naturally generate contigs of two haplotypes. Since our corrected reads contain allele information, we may use the SNP allele information to identify inconsistent overlaps. If the SNP alleles on a pair of reads are different, these two reads should come from different haplotypes and the overlaps between them should be inconsistent. At the error correction step, we have scored the possibility that two raw reads come from the same haplotype. However, the accuracy of the previous scoring that is based on the alignments between two raw reads cannot meet the requirements for identifying inconsistent overlaps. Therefore, we developed a read-level SNP caller and a read grouping method for identifying inconsistent overlaps.

**Fig. 2 | Performance comparison of error correction. a** Accuracy of raw and corrected reads and accuracy of SNP alleles in raw and corrected reads on the simulated datasets with different heterozygosity rates. **b** Accuracy of raw reads and corrected reads by NECAT and PECAT in difficult-to-map regions and low-complexity regions of HG002 reference genome. **c, d** Consistency, and completeness of raw reads and corrected reads by Canu, FALCON, MECAT2, NECAT, and PECAT on the seven diploid datasets. The metrics by Canu on *B. taurus* (PacBio CLR and ONT) and HG002 (ONT) and the metrics by FALCON on *B. taurus* (PacBio CLR) are excluded because they could not finish correcting in three weeks. Consistency is defined as $\sum \max(k_p, k_m) / \sum (k_p + k_m)$, in which $k_p$ and $k_m$ are the number of paternal and maternal haplotype-specific k-mers in each read. Completeness is the percentage of parent-specific k-mers (occurrences $\geq 4$) in the 40X longest reads. **e** Consistency of *D. melanogaster* (ISO1 × A4) raw reads and corrected reads by the different methods. Each point corresponds to a read. Its coordinate gives the proportion of the parental specific k-mers in the read, where k is 18. All 40X longest reads are shown in each sub-figure.

First, we map all corrected reads onto the haplotype-collapsed contigs from the first round of assembly using minimap2[35]. We call the heterozygous SNP sites based on the base frequency of the alignments (**Methods**). Then, we set each read as a template read and collect a set of reads that have the common SNP sites with it. We cluster the set of reads according to the SNP alleles in reads. The reads from the same haplotype are more likely to be clustered into the same subgroup. We then verify and correct the SNP alleles in the template read using other reads in the same subgroups. After identifying SNP alleles in each read, we remove the inconsistent overlaps using the SNP information from the candidate overlaps found in the first round of assembly. Then, we construct the string graph again. For high heterozygosity genome regions (>= 0.0005), this inconsistent overlap identification approach works well due to the high accuracy of SNPs in corrected reads. However, for low heterozygosity genome regions (< 0.0005), the SNP caller using corrected reads only can not achieve high performance in inconsistent overlap identification as the accuracy of SNPs is decreased dramatically in corrected reads for those genome regions. Therefore, we combine raw reads to identify and filter inconsistent overlaps. We first use corrected reads to identify inconsistent overlaps. Then, we call SNPs using raw reads and identify inconsistent overlaps again (**Methods**). For Nanopore reads, we have used Clair3[15] to call heterozygous SNP sites. For PacBio CLR reads, we do not have a tool to call SNPs from raw reads now.

We evaluate the performance of the inconsistent overlaps identification method using the haplotype reads classified by the trio-binning algorithm[16]. As shown in Supplementary Table 4, compared with the first round of assemblies, the percentages of inconsistent overlaps in simplified graphs of the second round of assemblies of diploid datasets *S. cerevisiae* (SK1×Y12), *A. thaliana* (Col-0×Cvi-0), *D. melanogaster* (ISO1×A4), *B. taurus* (Angus×Brahman), *A. thaliana* (Col-0×C24), *B. taurus* (Bison × Simmental) and HG002 decrease sharply. The percentages of inconsistent overlaps have decreased from 36.98%, 16.00%, 15.94%, 36.55%, 16.62%, 18.08%, and 37.45% in the first round of assemblies to 0.32%, 0.57%, 0.90%, 0.63%, 0.79%, 0.03%, and 1.09% in the second round of assemblies, respectively. For the three Nanopore datasets, we compare the performance of the method using corrected reads, raw reads, and both (Supplementary Table 4). Using both raw and corrected reads, we get lower percentages of inconsistent overlaps in simplified graphs of the second round of assemblies of all three Nanopore datasets. The high performance of the inconsistent overlap identification method ensures that PECAT can generate the haplotype-specific contigs effectively. After filtering inconsistent overlaps, collapsed regions in the first round of assembly are separated in the second round of assembly. The sizes of alternate contigs are close to their corresponding reference genome in the second round of assembly (Supplementary Table 5).

Moreover, our inconsistent overlap identification method can also help solve nearly identical repeats without knowing the number of their copies. The clustering step can automatically separate nearly identical repeats if there are SNPs that can divide the repeats. After filtering out the inconsistent overlaps at the repeats, PECAT can solve the repeat to generate contiguous contigs in the second round of assembly. As a result, PECAT achieves more contiguous assemblies in the second round of assembly on most of our diploid datasets

(Supplementary Table 5). As shown in Supplementary Figs. 3 and 4, PECAT perfectly solves repeats in the NCTC9024 and NCTC9006 datasets and reconstructs two circle contigs, while other assemblers obtain fragment results[7,37].

## Performance of PECAT error correction method

We evaluate the performance of the PECAT error correction method using four PacBio diploid datasets: *S. cerevisiae* (SK1×Y12), *A. thaliana* (Col-0×Cvi-0), *D. melanogaster* (ISO1×A4), *B. taurus* (Angus×Brahman), and three Nanopore diploid datasets: *A. thaliana* (Col-0×C24), *B. taurus* (Bison × Simmental), and HG002. We compare PECAT with the error correction methods in the other four tools, including Canu[5], FALCON[4], MECAT2[6], and NECAT[9] (Fig. 2 b–e and Supplementary Table 6). All methods have reported high accuracy (>98.6%). We also evaluate the performance of PECAT and NECAT on the difficult-to-map regions[38] and low-complexity regions of HG002 (**Methods**). On average, the accuracies of PECAT are 0.75% higher than those of NECAT, and the accuracy of corrected reads on these regions is similar to those on normal regions (Fig. 2b and Supplementary Table 7). We then assess the haplotype-specific k-mers completeness and consistency of corrected reads (**Methods**). These two metrics can evaluate the ability to retain consistent heterozygote alleles. The reads corrected by PECAT have higher completeness and consistency in all datasets (Fig. 2c, d). Especially, haplotype-specific k-mers consistencies of the reads corrected by PECAT are greater than or equal to 99.4% on all datasets, while haplotype-specific k-mers consistencies of the reads corrected by other methods are all less than or equal to 94.8%. The haplotype-specific k-mers completenesses of reads corrected by PECAT are also higher than those of reads corrected by other methods, especially for three Nanopore datasets. We plot the raw read and corrected reads using haplotype-specific k-mers for the seven datasets. As shown in Fig. 2e and Supplementary Figs. 5–10, PECAT effectively avoids mixing heterozygous alleles from two haplotypes and its corrected reads tend to contain only one type of haplotype-specific k-mer. In summary, the reads corrected by PECAT contain more haplotype-specific k-mers than those corrected by other methods.

## Performance of PECAT assembler

We also assess the performance of the PECAT assembler using four PacBio diploid datasets: *S. cerevisiae* (SK1×Y12), *A. thaliana* (Col-0×Cvi-0), *D. melanogaster* (ISO1×A4), *B. taurus* (Angus×Brahman), and three Nanopore diploid datasets: *A. thaliana* (Col-0×C24), *B. taurus* (Bison × Simmental) and HG002. We compare PECAT with four diploid genome assembly pipelines: Canu[5]+purge_dups[39], FALCON-Unzip[4], Flye[7]+HapDup[14,40], and Shasta[41] (Supplementary Note 2). We evaluate assembled genomes with respect to the assembly size, contiguity (contig NG50 and phase block NG50), and qualities (BUSCO[42], the base quality using pomosix (https://github.com/nanoporetech/pomoxis) and merqury[43], 'Intra-block switch error' from merqury, and the hamming error rate). (**Methods**).

For four PacBio CLR datasets, all pipelines output the contigs in primary/alternate format. As shown in Table 1, the sizes of both primary and alternate contigs of all four assemblies are close to those of their corresponding reference genome, except the alternate contigs of *S. cerevisiae* (SK1×Y12) genome assembled by Canu+Purge_dups.

**Table 1 | Performance comparison of assembly on PacBio CLR datasets**

| Dataset | Pipeline | Size (Mb) | NG50 (Mb) | Quality (reference-based) | Quality (k-mer-based) | BUSCO (%) | Hamming error (%) | Phase block NG50 (Mb) | Intra-block switch error (%) |
|---|---|---|---|---|---|---|---|---|---|
| *S. cerevisiae* SK1×Y12 200X | Ref | 12.1/12.0 | 0.9/0.9 | –/– | 47.3/49.4 | 99.6/99.6 | 0.13/0.03 | 0.9/0.9 | 0.13/0.03 |
| | Canu + Purge_dups | 12.4/4.8 | 0.8/0.0 | 25.6/30.5 | 38.9/36.2 | 98.7/28.1 | 39.24/6.25 | 0.0/0.0 | 9.92/4.50 |
| | FALCON-Unzip | 12.1/11.1 | 0.8/0.4 | 40.5/42.2 | 44.5/45.0 | 99.4/95.8 | 21.58/0.98 | 0.5/0.4 | 0.40/0.22 |
| | PECAT | 12.3/11.8 | 0.8/0.8 | 35.9/36.0 | 39.1/39.7 | 96.0/93.8 | 1.65/0.49 | 0.8/0.8 | 0.16/0.09 |
| *A. thaliana* Col-0×Cvi-0 164X | Ref | 133.3/119.7 | 26.2/23.20 | –/– | Inf/Inf | 99.3/99.2 | 0.12/0.01 | 12.1/23.2 | 0.11/0.01 |
| | Canu + Purge_dups | 129.1/122.6 | 6.7/0.1 | 28.0/30.4 | 24.8/30.7 | 98.7/81.7 | 37.52/2.65 | 0.1/0.1 | 2.81/2.45 |
| | FALCON-Unzip* | 140.0/104.9 | 8.0/4.3 | 40.0/40.0 | 30.0/34.9 | 98.9/93.6 | 15.11/0.98 | 3.1/2.4 | 0.15/0.19 |
| | PECAT | 130.6/120.4 | 14.3/7.8 | 34.6/34.7 | 25.3/33.2 | 98.3/98.2 | 2.64/0.21 | 12.6/7.8 | 0.13/0.13 |
| *D. melanogaster* ISO1×A4 200X | Ref | 143.7/140.7 | 25.3/25.0 | –/– | 46.5/46.3 | 98.7/98.7 | 0.06/0.02 | 25.3/24.6 | 0.06/0.02 |
| | Canu + Purge_dups* | 143.2/129.7 | 16.1/0.3 | 33.6/34.7 | 35.4/35.3 | 98.5/87.3 | 43.50/3.97 | 0.4/0.3 | 3.92/3.01 |
| | FALCON-Unzip | 189.7/105.9 | 4.0/0.4 | 40.5/40.5 | 37.1/37.4 | 98.8/84.0 | 29.31/5.38 | 1.0/0.3 | 0.34/0.41 |
| | PECAT | 149.6/135.7 | 24.5/11.9 | 38.5/39.2 | 40.8/43.5 | 98.7/96.7 | 3.00/0.07 | 16.1/11.8 | 0.05/0.04 |
| *B. taurus* Angus×Brahman 135X | Ref | 2580.8/2681.0 | 911.1/104.5 | –/– | 43.3/43.8 | 93.6/95.6 | 0.10/0.04 | 21.8/30.5 | 0.08/0.03 |
| | FALCON-Unzip* | 2713.4/2453.7 | 31.4/2.0 | 39.2/38.5 | 39.4/39.0 | 95.4/86.3 | 28.15/1.97 | 3.2/1.8 | 0.21/0.22 |
| | PECAT | 2744.7/2447.6 | 72.4/2.8 | 34.6/34.6 | 39.9/40.1 | 94.8/87.0 | 29.46/0.47 | 4.5/2.4 | 0.10/0.09 |

'Size' is the total number of base pairs in all contigs generated by assemblers. 'NG50' is the length of the shortest contig for which longer and equal length contigs cover at least 50 of genome size. The genome sizes of *S. cerevisiae*, *A. thaliana*, *D. melanogaster*, and *B. taurus* that we used for evaluation are 12 M, 130 M, 140 M, and 2.7 G, respectively. 'BUSCO' is gene completeness evaluated by BUSCO. 'Quality (reference-based)' is the metric 'qS0' evaluated by mercury. 'Hamming error' is the fraction of nondominant parental-specific k-mers in a contig. 'Quality (k-mer-based)', 'Phase block NG50', and 'Intra-block switch error' are evaluated by mercury. All assemblies are in primary/alternate format. The primary and alternate contigs are separately reported in each cell. 'Ref' is the reference genome. The sources of the reference genomes are illustrated in Supplementary Table 17. For *B. taurus*, Canu didn't finish the assembly in 3 weeks, so it is excluded. Asterisks mark previously published assemblies.

Compared to the other two pipelines, PECAT obtains the highest NG50 with 0.8/0.8, 14.3/7.8, 24.5/11.9, and 72.4/2.8 Mb and the highest 'phase block NG50' with 0.8/0.8, 12.1/7.4, 16.1/11.8, 4.5/2.4 Mb for all four assemblies. As shown in Supplementary Figs. 11–14, the alternate contigs generated by all pipelines are haplotigs. Unlike the other two pipelines, most primary contigs by PECAT are also haplotigs except those of *B. taurus* (Angus×Brahman). These results are consistent with the results of the 'Hamming error', where the 'Hamming error' of primary contigs of *B. taurus* (Angus×Brahman) is exceptionally high. All pipelines have similar 'Quality' and BUSCO scores, except that Canu+Purge_dups has low BUSCO scores on alternate contigs of *S. cerevisiae*.

For three Nanopore datasets, Canu+Purge_dups and Shasta output the contigs in primary/alternate format, Flye+HapDup outputs the contigs in dual assembly format and PECAT can output both formats. As shown in Table 2, the sizes of both two sets of contigs of all three assemblies are close to their corresponding reference genome, except the *A. thaliana* (Col-0×C24) genomes assembled by Canu+Purge_dups. Compared to other pipelines, PECAT obtains higher NG50 for three genomes. The phase block NG50 reported by PECAT is at least 10 times higher than those reported by other pipelines for *A. thaliana* (Col-0×C24) and *B.taurus* (Bison×Simmental). Especially, PECAT reports the assembly with phase block NG50 of 79.6/86.1 Mb for *B. taurus* (Bison×Simmental), which exceeds the assembly with phase block N50 of 68.5/70.6 Mb reported by the trio-binning method using additional parental reads[44]. Meanwhile, for HG002, PECAT reports higher phase block NG50 than that reported by Flye+HapDup, both are at least 25 times higher than that reported by Shasta. PECAT reports smaller Intra-block switch error and Hamming error on *A. thaliana* (Col-0×C24) and *B. taurus* (Bison×Simmental). Most of the contigs reported by PECAT on *A. thaliana* (Col-0×C24) and *B. taurus* (Bison × Simmental) are haplotigs (Fig. 3a and Supplementary Fig. 15). For HG002, PECAT and Flye+HapDup report similar Intra-block switch errors, and both are much less than those reported by Shasta. Although PECAT and Flye+HapDup reported smaller Hamming errors than that reported by Shasta, their reported Hamming error rates are high, which may be because some low heterozygosity regions of HG002 are not successfully phased. Moreover, most contigs reported by Flye+HapDup and PECAT on HG002 are not haplotigs (Supplementary Figs. 16,17).

For the metrics 'Quality' and BUSCO score, Shasta reports the lowest scores on all three Nanopore datasets, while Flye+HapDup and PECAT report similar scores on *A. thaliana* (Col-0×C24), *B. taurus* (Bison×Simmental) and HG002, except that the BUSCO scores of the alternate contigs/the haplotype 2 contigs reported by PECAT are slightly lower than those reported by Flye+HapDup (91.0%/91.6% vs. 94.1%). The main reason for this difference is that PECAT places the contigs of X and Y chromosomes in the primary contigs or the haplotype 1 contigs while Flye+HapDup outputs two copies of contigs of X and Y chromosomes to two sets of assemblies at the same time. We remove the contigs of X and Y chromosomes in the assemblies of Flye+HapDup (dual) and PECAT (dual). The BUSCO scores of the assembly by Flye+HapDup (dual) are reduced from 94.2%/94.1% to 91.2%/91.1%, while the BUSCO scores of the assembly by PECAT (dual) are reduced from 94.6%/91.6% to 91.6%/91.5%, which are similar to those reported by Flye+HapDup (dual).

We also compare the HG002 genome assembled using Nanopore reads by PECAT with those using HiFi reads by Hifiasm[25,27]. The long Nanopore reads can help the assemblers to obtain longer phased blocks. PECAT reports much longer NG50 and phase block NG50 than those reported by Hifiasm using HiFi reads only, even longer than those reported by Hifiasm using both HiFi reads and Hi-C reads. The Intra-block switch errors reported by PECAT are similar to those reported by Hifiasm. On the other hand, the high-quality HiFi reads allow Hifiasm to report higher base qualities of assemblies with better

**Table 2 | Performance comparison of assembly on Nanopore and PacBio HiFi datasets**

| Dataset | Pipeline | Size (Mb) | NG50 (Mb) | Quality (reference-based) | Quality (k-mer-based) | BUSCO (%) | Hamming error (%) | Phase block NG50 (Mb) | Intra-block switch error (%) |
|---|---|---|---|---|---|---|---|---|---|
| *A. thaliana* Col-0 × C24 ONT, 106X | Ref | 133.3/119.2 | 26.2/23.8 | –/– | 44.7/44.0 | 99.3/99.1 | 0.14/0.01 | 12.7/23.8 | 0.15/0.02 |
| | Canu + Purge_dups (pri/alt) | 125.9/68.3 | 8.1/0.0 | 22.0/23.5 | 29.8/28.7 | 96.6/37.4 | 37.95/7.02 | 0.1/0.0 | 8.33/5.71 |
| | Flye + HapDup (dual) | 136.4/136.5 | 2.1/2.1 | 25.6/25.0 | 28.5/28.3 | 97.8/97.8 | 11.85/13.62 | 0.5/0.5 | 0.66/0.67 |
| | Shasta (pri/alt) | 126.3/105.0 | 0.8/0.5 | 22.4/23.8 | 28.9/30.6 | 94.9/92.2 | 11.75/7.96 | 0.3/0.3 | 7.11/5.39 |
| | PECAT (pri/alt) | 131.2/123.9 | 14.3/7.7 | 30.2/30.6 | 31.4/32.1 | 98.5/98.2 | 3.18/0.32 | 9.1/7.1 | 0.35/0.36 |
| | PECAT (dual) | 131.1/125.1 | 14.3/14.0 | 30.4/30.7 | 31.6/32.2 | 98.6/98.4 | 1.07/0.38 | 7.8/7.9 | 0.37/0.37 |
| *B. taurus* Bison × Simmental ONT, UL, 200X | Ref | 2651.6/2861.7 | 87.8/104.4 | –/– | 38.7/36.1 | 93.1/95.7 | 0.67/0.48 | 87.8/104.4 | 0.74/0.61 |
| | Flye + HapDup (dual) | 2713.0/2713.2 | 33.4/33.4 | 25.0/25.0 | 28.5/28.4 | 83.4/83.3 | 6.52/6.57 | 8.2/8.1 | 0.44/0.44 |
| | Shasta (pri/alt) | 2815.7/2406.9 | 0.3/0.2 | 15.1/15.1 | 22.9/24.0 | 61.3/59.1 | 22.05/25.65 | 0.1/0.1 | 23.26/25.79 |
| | PECAT (pri/alt) | 2968.2/2770.0 | 94.3/93.8 | 25.6/25.6 | 28.8/28.5 | 83.6/83.1 | 0.39/0.38 | 79.6/86.1 | 0.37/0.38 |
| | PECAT (dual) | 2970.2/2856.3 | 94.3/93.8 | 25.5/25.6 | 28.8/28.8 | 83.3/83.4 | 0.39/0.38 | 79.5/86.1 | 0.37/0.39 |
| HG002 ONT, R9, UL 59X | Ref | 2959.3/3061.7 | 146.1/154.4 | –/– | 58.6/59.4 | 92.8/95.8 | 0.15/0.08 | 90.4/106.7 | 0.02/0.03 |
| | Flye+ HapDup (dual) | 2932.2/2932.4 | 49.9/49.9 | 33.2/33.2 | 40.9/40.8 | 94.2/94.1 | 7.91/8.24 | 20.5/19.6 | 0.08/0.09 |
| | Shasta (pri/alt) | 3007.0/2443.9 | 24.2/2.9 | 28.8/30.1 | 34.7/38.5 | 92.4/79.6 | 18.93/11.42 | 0.8/0.5 | 8.66/5.36 |
| | PECAT (pri/alt) | 3059.3/2857.7 | 92.9/15.0 | 30.8/30.8 | 41.8/41.6 | 94.7/91.0 | 15.72/1.48 | 22.2/13.0 | 0.08/0.11 |
| | PECAT (dual) | 3057.2/2927.2 | 92.7/74.5 | 30.8/30.8 | 41.8/41.8 | 94.6/91.6 | 9.67/10.98 | 30.8/23.6 | 0.08/0.11 |
| HG002 ONT, R10, UL, 116X | Flye+ HapDup (dual) | 2954.7/2952.1 | 61.9/59.3 | 36.4/36.2 | 48.9/48.2 | 95.8/95.7 | 3.13/3.02 | 46.5/39.1 | 0.02/0.03 |
| | Shasta (pri/alt) | 3095.4/2729.9 | 45.3/33.8 | 34.3/34.4 | 42.1/48.7 | 95.7/92.2 | 3.05/0.86 | 17.2/15.4 | 0.30/0.44 |
| | PECAT (pri/alt) | 3159.7/2895.6 | 91.4/59.9 | 38.0/38.0 | 49.0/50.1 | 95.8/92.4 | 3.75/0.30 | 59.4/58.0 | 0.04/0.05 |
| | PECAT (dual) | 3153.4/2917.8 | 91.4/80.2 | 38.0/38.0 | 49.2/50.0 | 95.8/92.7 | 2.99/1.49 | 63.8/59.2 | 0.04/0.06 |
| HG002 HiFi, 36X | Hifiasm (pri/alt) * | 3112.2/2910.3 | 89.9/0.4 | 50.0/50.0 | 55.3/56.5 | 95.6/77.0 | 24.38/1.67 | 1.1/0.3 | 0.11/0.01 |
| | Hifiasm (dual) * | 3015.3/3077.5 | 44.8/64.5 | 50.0/50.0 | 54.9/57.8 | 95.5/94.9 | 34.80/24.10 | 1.0/1.0 | 0.16/0.11 |
| | Hifiasm (Hi-C) * | 3075.0/2908.7 | 55.1/55.1 | 50.0/50.0 | 54.9/57.9 | 95.7/92.5 | 0.45/0.25 | 20.6/20.4 | 0.09/0.06 |

'Size' is the total number of base pairs in all contigs generated by assemblers. 'NG50' is the length of the shortest contig for which longer and equal length contigs cover at least 50 of genome size. The genome sizes of *A. thaliana*, *B. taurus*, and HG002 that we used for evaluation are 130 M, 2.7 G, and 3 G, respectively. 'BUSCO' is gene completeness evaluated by BUSCO. 'Hamming error' is the fraction of nondominant parental-specific k-mers in a contig. 'Quality (reference-based)' is the metric 'q50' evaluated by Pomoxis. 'Quality (k-mer-based)', 'Phase block NG50', and 'Intra-block switch error' are evaluated by mercury. 'pri/alt' represents primary/alternate assembly format. 'dual' represents dual assembly format. 'ONT' indicates the dataset is composed of Nanopore reads. 'UL' indicates the reads are ultra-long reads. 'HiFi' indicates the dataset is composed of PacBio HiFi reads. 'Hi-C' represents that the assembly uses the additional Hi-C reads. The two sets of contigs are separately reported in each cell. 'Ref' is the reference genome. The sources of the reference genomes are illustrated in Supplementary Table 17. For *B. taurus* and HG002, Canu didn't finish the assembly in 3 weeks, so it is excluded. Asterisks mark previously published assemblies.

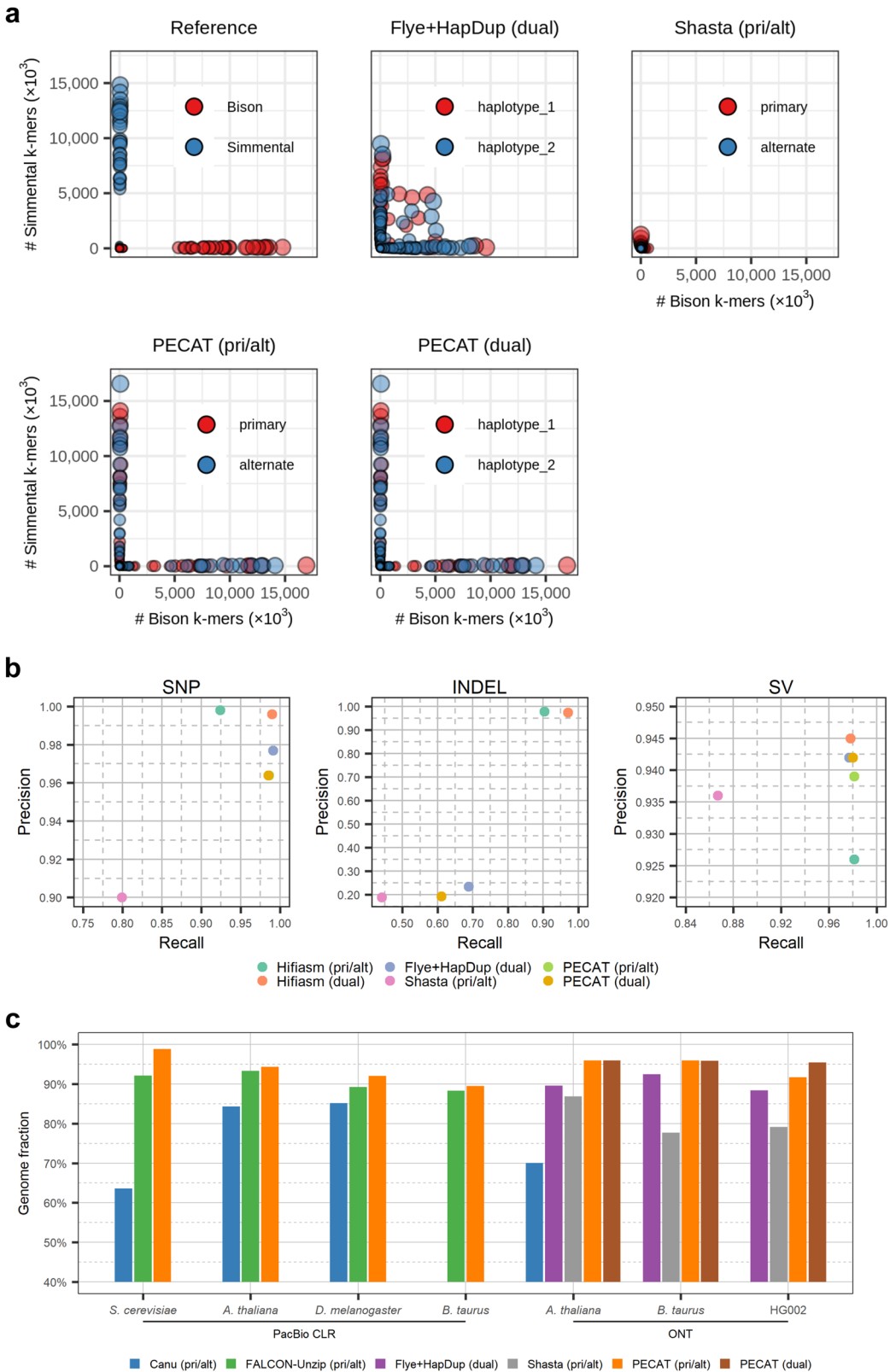

**Fig. 3 | Performance comparison of assembly. a** Haplotype-specific k-mer blob plots of the *B. taurus* (Bison × Simmental) reference genome and assemblies by Flye +HapDup, Shasta, and PECAT. pri/alt or dual represents that the assembly is the primary/alternate format or the dual assembly format. Each blob corresponds to a contig. The coordinate of the blob gives the count of the parental specific k-mers in the contig, where k is 21. Blob size is proportional to contig length. **b** Precisions and recalls of small variants (SNP, INDEL) and structural variants (SV) in HG002 assemblies. **c** Genome fraction of the assemblies, which are evaluated by QUAST.

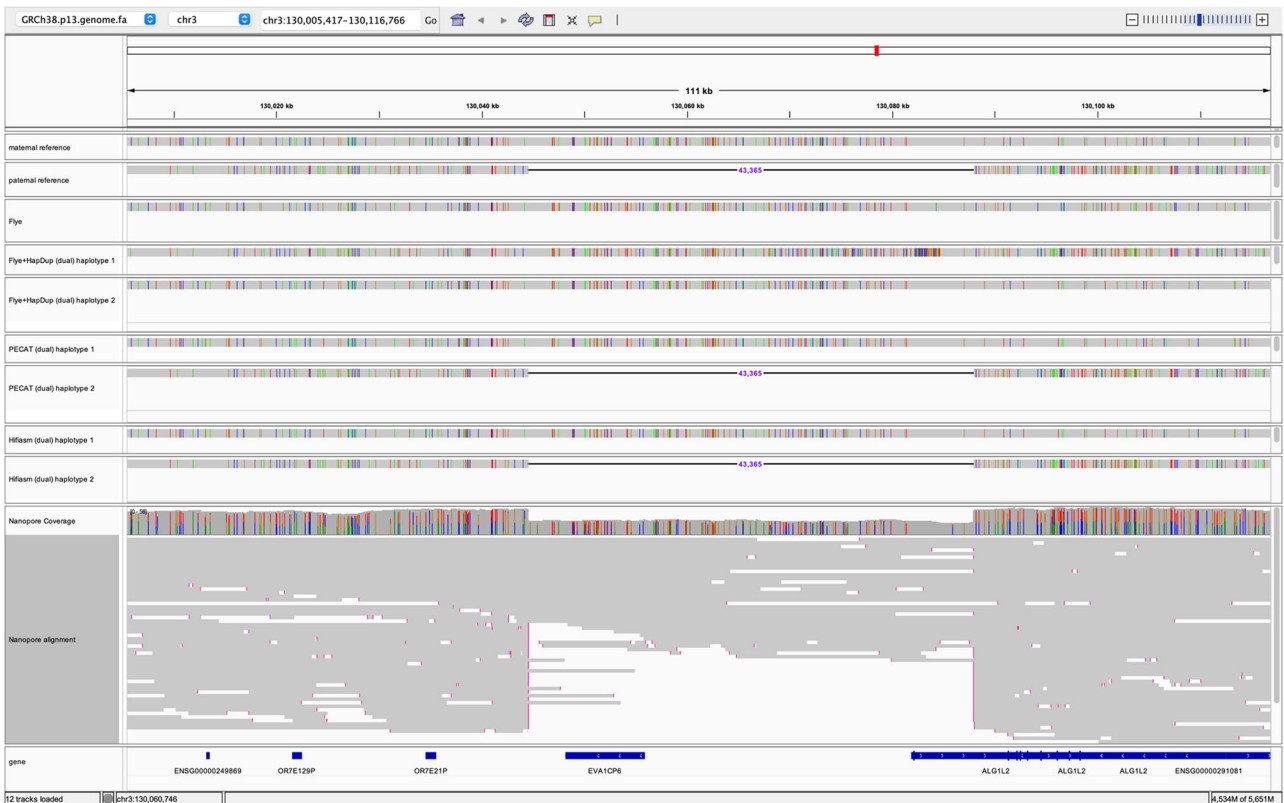

**Fig. 4 | Screenshot of HG002 reference, assembly and read alignment to GRCh38.** It shows the range of [130,005,418, 130,116,766] in chromosome 3. Small INDELs in all alignments and mismatches in read alignments are not shown. Paternal reference and maternal reference are HG002 paternal references. The assemblies by Flye, Flye+HapDup, and PECAT are from Nanopore reads. The assembly by Hifiasm is from HiFi reads. The assembly by Flye is haplotype-collapsed contigs. The other assemblies are in the dual assembly format. The two sets of contigs are labeled as haplotype 1 and haplotype 2.

'Quality' scores. Moreover, Hifiasm reports higher Hamming errors when using HiFi reads only. However, with the help of Hi-C data, Hifiasm reported a much smaller Hamming error and almost all contigs are haplotigs (Supplementary Figs. 16,17). We then evaluate the small variants (SNP, INDEL) and the structural variants (SV) in HG002 assemblies against the GIAB benchmark (**Methods**). As shown in Fig. 3b and Supplementary Table 8, the precisions and recalls of SNP and SV of genomes assembled by Flye+HapDup and PECAT using Nanopore reads are similar to those of genomes assembled by Hifiasm using HiFi reads. However, the precisions and recalls of INDEL of genomes assembled by Flye+HapDup and PECAT using Nanopore read is much less than those of genomes assembled by Hifiasm using HiFi reads, which is consistent with the previous finding[14].

In the second round of assembly of PECAT, it reassembles the filtered string graph, which will help correct errors or collapsed regions in the first round of assembly. As shown in Fig. 3c and Supplementary Data 1, PECAT reports higher 'Genome fraction' values than those reported by other pipelines. We then map the HG002 reference genome and HG002 assemblies to GRCh38. As an example (Fig. 4), the region [130,005,418, 130,116,766] in chromosome 3 includes the gene EVA1CP6 and has a large INDEL with a length of about 43,365 bp between the parental reference genomes. Both PECAT and Hifiasm reconstruct the INDEL in their assemblies, but Flye +HapDup (dual) fails. This large INDEL cannot be preserved in the haplotype-collapsed contigs in the first round of assembly and reads from another haplotype may not be mapped onto the haplotype-collapsed contigs. Therefore, it is difficult for the polish-based assemblies, such as Flye+HapDup, to restore the INDEL in the second round of assembly, which may be the reason that Flye+Hapdup reports

lower Genome fraction scores than those reported by PECAT (Fig. 3c and Supplementary Data 1) on three Nanopore datasets.

We further compare the computational resources required by each pipeline (Supplementary Table 9, 10). Canu on *B. taurus* (Angus×Brahman), *B. taurus* (Bison×Simmental), and HG002, FALCON-Unzip on *B. taurus* (Angus×Brahman) are excluded for comparison since the assemblies are not constructed within three weeks. PECAT is at least 8.7 times faster than traditional correct-then-assemble pipelines, such as Canu+purge_dups and FALCON-Unzip. But it's slower than assemble-then-correct pipelines, such as Shasta and Flye+HapDup. The peaks of memory usage and disk space usage of PECAT are also recorded in Supplementary Table 10.

**Performance on highly accurate long reads**

With the development of sequencing technology, the accuracy of long reads has greatly improved recently[24,45]. We evaluate the performance of PECAT using Nanopore R10 sequencing (ultra-long), Nanopore R10 duplex sequencing and PacBio HiFi sequencing reads. The accuracy of 40X longest reads for those datasets is 98.25%, 99.67%, and 99.77% (Supplementary Fig. 18 and Supplementary Table 11), respectively, which are much higher than the accuracy of the Nanopore R9 (ultra-long) dataset (94.97%). However, there are still less accurate reads in those datasets and PECAT error correction can still improve the qualities of those reads, except for a 0.1–0.2% decrease in the metric "Completeness" (Supplementary Table 11). The Nanopore R10 (ultra-long) dataset with higher accuracy and longer read length improves the assemblies of all pipelines (Table 2). Compared with the assemblies from the Nanopore R9 (ultra-long) dataset (Table 2), the assemblies from the Nanopore R10 (ultra-long) dataset have higher QV when

evaluated by pomoxis and merqury. Especially, the assembly of PECAT from the Nanopore R10 (ultra-long) dataset has the highest phase block NG50 with 59.4/58.0 Mb in primary/alternate format and with 63.8/59.2 Mb in dual format, which is twice as much as those of assembly of PECAT from Nanopore R9 (ultra-long) data. Furthermore, the assemblies from the Nanopore R10 (ultra-long) dataset report one magnitude smaller Hamming error and more contigs in this assembly are haplotigs (Supplementary Figs. 19, 20). Similar to the Nanopore R9 (ultra-long) dataset, the assembly of PECAT from the Nanopore R10 (ultra-long) dataset outperforms those of Flye+HapDup and Shasta in terms of NG50 and phase block NG50. Meanwhile, the assemblies for the Nanopore R10 duplex and PacBio HiFi datasets (Supplementary Table 12) have less NG50 and phase block NG50 than those of assembly from the Nanopore R9 (ultra-long) dataset (Table 2), although have higher quality measures. Moreover, less amount of contigs of assemblies of Nanopore R10 duplex and PacBio HiFi reads are haplotigs (Supplementary Figs. 21–24). This result indicated that the length of reads is more important for obtaining contiguous haplotype-specific contigs.

We then evaluate the small variants (SNP, INDEL) and the structural variants (SV) in those assemblies using Nanopore R10 sequencing (ultra-long), Nanopore R10 duplex sequencing and PacBio HiFi sequencing reads against the GIAB benchmark (Supplementary Table 13–15 and Supplementary Figs. 25–27). The assemblers report similar metrics. The precisions and recalls of SNP and SV of the assemblies are similar to those of the assemblies using Nanopore R9 (ultra-long) reads (Fig. 3b and Supplementary Table 8). However, the precisions and recalls of INDEL of genomes assembled from Nanopore R10 (ultra-long) reads and Nanopore R10 duplex reads are much greater than those of assemblies using R9 (ultra-long) reads, while still less than those of assemblies using HiFi reads.

## Discussion

Although long noisy reads, especially Nanopore reads, have the advantage to generate high contiguity contigs, using them for diploid genome assembly remains a challenge. Due to the high sequencing error rate, directly phasing the raw reads and then assembling the diploid genome can not obtain high-quality assembly[46]. Here, we first develop a haplotype-aware error-correct method to keep most of the heterozygote alleles while correcting sequencing errors. However, even after error corrections, it is not able to accurately call SNPs by just aligning corrected reads as the Hifiasm did for HiFi reads. PECAT then first generates a haplotype-collapsed genome and calls SNPs by aligning the corrected reads to the haplotype-collapsed genome. Nevertheless, the accuracy of this SNP call is still low for low heterozygosity rate regions. Therefore, we use Clair3 to call SNPs again for Nanopore reads. Our experiments show that the SNPs called from raw reads complemented the SNPs called by aligning corrected reads. Combing both SNPs helps achieve the best performance.

Another advantage of our haplotype-aware error-corrected reads is to allow reuse the overlaps built in the first round of assembly and just simplify the overlaps by removing inconsistent ones before reassembling the diploid genome. This strategy is like that used in Hifiasm. Compared to methods that directly separate the collapsed contigs into two haplotypes, such as Flye+HapDup, this strategy can correct errors or collapsed regions in the first round of assembly, and then achieve better assembly.

Furthermore, PECAT does not simply phase the reads into two copies, but uses the read grouping method to separate reads from nearly identical repeats into multiple copies, which can help solve the repeats with more than two copies. Meanwhile, in order to obtain high-quality phased assembly, PECAT needs higher coverage of data. As shown in Supplementary Table 16, the contiguity and qualities of the assembly of HG002 using 37X Nanopore reads are less than those of assembly using 59X Nanopore reads.

Although the new generation long reads from Nanopore and PacBio have much higher accuracy, PECAT error correction can still improve their quality. PECAT can also efficiently assemble those highly accurate reads. It can leverage the advantages of read length and accuracy to obtain better assemblies. Therefore, PECAT achieve the phase block NG50 with 59.4/58.0 Mb in primary/alternate format and with 63.8/59.2 Mb in dual format only using Nanopore R10 (ultra-long) reads. However, our current error correction method may not be effective enough to distinguish small errors from heterozygote alleles in very low heterozygosity rate regions, even in highly accurate reads (Nanopore duplex and PacBio HiFi reads). Compared with the read-length advantage, PECAT does not fully take advantage of the high accuracy of reads for being compatible with long noisy reads, therefore it achieves mediocre performance on Nanopore duplex and PacBio HiFi reads (Supplementary Table 12 and Supplementary Figs. 21–24). We will resolve this issue in subsequent PECAT releases. Overall, PECAT is an efficient assembly pipeline for diploid genomes.

## Methods

### Diploid datasets for benchmarking

We evaluate the performance of PECAT using seven diploid datasets (Supplementary Note 1 and Supplementary Table 17): *S. cerevisiae* (SK1×Y12), *A. thaliana* (Col-0×Cvi-0), *D. melanogaster* (ISO1×A4), *B. taurus* (Angus×Brahman), *A. thaliana* (Col-0×C24), *B. taurus* (Bison×Simmental) and HG002. The first four datasets are PacBio CLR datasets and the last three are Nanopore datasets (the last two are ultra-long raw reads and the HG002 dataset is generated by Nanopore R9). The heterozygosity rate of those species is 0.85%, 1.04%, 0.84%, 1.12%, 0.83%, 1.48%, and 0.34%, respectively (Supplementary Table 1 and Supplementary Note 3). The accuracies of reads in those datasets are 87.80%, 88.24%, 89.58%, 86.25%, 92.33%, 89.12%, and 94.97%, respectively. (Supplementary Table 6). In addition, we evaluate the performance of PECA on more accurate datasets, i.e., other three HG002 datasets using Nanopore R10 sequencing (ultra-long), Nanopore R10 duplex sequencing and PacBio HiFi sequencing, separately. Their accuracies are 98.25%, 99.67%, and 99.77% (Supplementary Table 11). For each of the above data, PECAT and NECAT extract the longest 80X raw reads or all reads, if the dataset is less than 80X, for error correction and assembly. Canu corrects the longest 100X raw reads for assembly. Other tools use all raw reads for error correction or assembly (Supplementary Note 2).

### Haplotype-aware error correction

The PECAT error correction method is based on the partial-order alignment (POA) graph method[4,47]. Instead of assigning the same weight to each read for error correction, we select reads more likely coming from the same haplotype or the same copy of the nearly identical repeat and assign different weights to prevent heterozygotes in reads from being eliminated as sequencing errors. First, we find candidate overlaps between raw reads using minimap2 with parameters "-x ava-pb" for PacBio CLR reads and parameters "-x ava-ont" for Nanopore reads. For each template read to be corrected, we collect a group of supporting reads that have candidate overlaps with the template read. Then, we perform pairwise alignment between the template read and each supporting read using diff[36] or edlib[48] algorithm and build a POA graph based on the alignments of the reads as shown in Supplementary Fig. 1. Each node in the graph is labeled by a triple value $(c,r,b)$, corresponding a base pair in the alignment. $c$ is the location at the template read, $r$ means the number of consecutive insertions (if $r = 0$, there is a match, a mismatch, or a deletion), and $b$ is the base on the supporting read or a deletion, which is one of $\{'A','C','G','T','-'\}$. Each edge in the graph means two base pairs of their nodes appearing continuously in the alignment. The support count of each edge is defined as the number of alignments which pass the edge.

For the convenience of analysis, we add a trivial alignment between the template read and itself to the graph. According to our observations, for each location $c$, all paths in the graph must only pass one of the nodes $\{(c,0,b_i),b_i \in ('A','C','G','T','-')\}$. If these nodes have more than one in-edge with large support counts, there may be heterozygotes rather than random sequencing errors (Supplementary Fig. 1 b). Therefore, we compute the support count $s_i$ for in-edges of the nodes. We mark an in-edge as the important one if

$$s_i \geq \begin{cases} r_l \cdot S & S \leq S_l \\ \frac{r_h \cdot S_h - r_l \cdot S_l}{S_h - S_l} \cdot (S - S_l) + r_l \cdot S_l & S_l < S \leq S_h \\ r_h \cdot S & S_h < S \end{cases}, \text{ where } S = \sum s_i, \text{ and } r_l, r_h,$$

$S_l$, and $S_h$ are user-set parameters (the default values are 0.5, 0.2, 10, and 200). We mark a location $c$ as the important one if (1) there is no homopolymer at location $c$, (2) there are two or more important in-edges and a bubble structure which can be detected along the reverse direction of the in-edges, (3) more than half of the reads through the important in-edge pass the corresponding path in the bubble, (4) there is not an INDEL variant between the sequences represented by the paths in the bubble. Then, we calculate a score for each supporting read based on whether the supporting read and the template read pass the same important in-edges at important locations. We increase the score by 1 if a supporting read passes the same important in-edge as template read and decrease it by 1 if the supporting read and the template read pass the different important in-edges (Supplementary Fig. 1 c). For uniformity, the score is divided by the number of important locations in the supporting read. As shown in Supplementary Fig. 1 d, the histogram of scores shows two or more peaks if the supporting reads come from different haplotypes or different copies of the segmental duplication. We select the reads whose scores fall into the first peak for error correction. If there is only one peak, we select half of the supporting reads with larger scores. We linearly map the scores of selected reads to a range, which is [0.4,0.8] by default, as the final weights of supporting reads. The weight of each edge is the sum of the weights of selected supporting reads that pass the edge.

Finally, we find the path in the POA graph to generate a consensus for the template read. For each node $v$ labeled by the triple value $(c,r,b)$, if it has $N$ in-edges $\{(u_i,v)|i \leq N\}$, it gets the score $S_v = \max_{i \leq N}\{S_{u_i} + W_{(u_i,v)} - P_c\}$, where $W_{(u_i,v)}$ is the weight of the edge $(u_i,v)$. In the previous work[47], $P_c$ is the half of the coverage at the location $c$ in the template read. In our work, $P_c = \max(0.4^r,0.3)*W_c$ instead, where $W_c$ is the sum of weight of the supporting reads which pass the location $c$. The score of the nodes without any in-edge is assigned to 0. We calculate the scores for all nodes by dynamic programming in topological order and record the related edges which get the maximum scores for the nodes. The node with the highest score is selected and backtracking is done to obtain the path for consensus. The consensus sequence for the template read is generated by concatenating the bases of the nodes in the path.

## Read-level SNP caller and read grouping method for identifying inconsistent overlaps

**Calling heterozygous SNPs in haplotype-collapsed contigs and SNP alleles in reads.** We map corrected reads to the first round of assembly using minimap2 with parameters "-x map-pb -c -p 0.5 -r 1000" for PacBio CLR reads and parameters "-x map-ont -c -p 0.5 -r 1000" for Nanopore reads. It performs base-level alignment and generates CIGAR strings. We scan the CIGAR strings and call heterozygous SNPs for each contig. We call a base site a heterozygous SNP site if it meets the following two conditions. (1) Its coverage is in the range [10, 1000]. (2) The number of second-most common bases is greater or equal to

$$\begin{cases} r_l \cdot c & c \leq c_l \\ \frac{r_h \cdot c_h - r_l \cdot c_l}{c_h - c_l} \cdot (c - c_l) + r_l \cdot c_l & c_l < c \leq c_h \\ r_h \cdot c & c \geq c_h \end{cases}, \text{ in which } c \text{ is the site coverage}$$

and $r_l, r_h, c_l$, and $c_h$ are user-set parameters (default values are 0.4, 0.2, 10 and 100). After calling heterozygous SNPs, we call SNP alleles in reads. We define a function $H_r(s)$ for each read $r$. For any SNP site $s$, $H_r(s)$ is defined as 1 or 2 if the read $r$ covers the site $s$ and the base of the read $r$ is equal to the first-most or second-most common base at site $s$. Otherwise, $H_r(s)$ is defined as 0.

**Verifying and correcting SNP alleles.** To verify and correct SNP alleles in a read, we need to find which reads are from the same haplotype within its local region. For each read labeled as the template read $t$, we assume that it covers a set of SNP sites $S = \{s_1, s_2, \ldots, s_N\}$. We collect the query reads that cover 3 common SNP sites with the template read $t$. The template read and the query reads are put into a group $G$. We cluster the reads in the group $G$ according to the SNP alleles in them. The reads in the same cluster can be considered from the same haplotype. To facilitate the description of the method, here we make some definitions. We define a centroid $C = \{(v_s, n_s)|s \in S\}$ for the group $G$ at the SNP site set $S$. $v_s$ is defined to $v_s^+ - v_s^-$, where $v_s^+$ is the number of the read set $\{r|H_r(s) = 1, r \in G, s \in S\}$ and $v_s^-$ is the number of the read set $\{r|H_r(s) = 2, r \in G, s \in S\}$. $n_i$ is defined to $v_s^+ + v_s^-$. One read can be regarded as a group only including it. We define the following three formulas to get verified SNP sites of the centroid $C$.

$$V_>(C,p_0,p_1) = \{s|v_s > 0, |v_s| \geq \max(p_0 \cdot n_s, p_1)\}, (v_s, n_s) \in C\}$$
$$V_<(C,p_0,p_1) = \{s|v_s < 0, |v_s| \geq \max(p_0 \cdot n_s, p_1)\}, (v_s, n_s) \in C\}$$
$$V_{\neq}(C,p_0,p_1) = V_>(C,p_0,p_1) \bigcup V_<(C,p_0,p_1)$$

The parameters $p_0$ and $p_1$ are used to control which sites are valid. Next, we define the operations between two centroids.

$$A(C_1,C_2,p_0,p_1) = V_{\neq}(C_1,p_0,p_1) \bigcap V_{\neq}(C_2,p_0,p_1)$$
$$S(C_1,C_2,p_0,p_1) = V_<(C_1,p_0,p_1) \bigcap V_<(C_2,p_0,p_1) \cup V_>(C_1,p_0,p_1) \bigcap V_>(C_2,p_0,p_1)$$
$$D(C_1,C_2,p_0,p_1) = V_<(C_1,p_0,p_1) \bigcap V_>(C_2,p_0,p_1) \cup V_>(C_1,p_0,p_1) \bigcap V_<(C_2,p_0,p_1)$$

$A(C_1,C_2,p_0,p_1)$ is the set of common verified SNP sites of the centroids $C_1$ and $C_2$. $S(C_1,C_2,p_0,p_1)$ and $D(C_1,C_2,p_0,p_1)$ are used to describe the similarity and the distance of the centroids $C_1$ and $C_2$. A read $r$ can also be used as the parameter, which means the centroid of $\{r\}$.

For each template read $t$ and related group $G$, we use a divide-then-combine strategy to cluster reads (Supplementary Fig. 28). In the dividing step, we use a modified bisecting k-means algorithm[49] to divide the group $G$ into small ones with following steps

1. The centroids $C_1$ and $C_2$ of the sets $\{r_1\}$ and $\{r_2\}$ are initially selected, where reads $r_1$ and $r_2$ are a read pair in the group $G$ with the largest distance ($r_1, r_2 = \arg_{r_1, r_2 \in G} \max |D(r_1, r_2, 0, 0)|$).

2. We divide the group $G$ into three subgroups, two subgroups corresponding to the centroids and a separate subgroup containing the reads far away from the centroids. For each read $r$, if it is far away from the centroids, it meets the conditions: $\frac{|A(r,C_i,0,0)|}{|V_{\neq}(r,0,0)|} < p_2$ or $\frac{|D(r,C_i,0,0)|}{|A(r,C_i,0,0)|} > p_3, i \in \{1,2\}$, where $p_2$ and $p_3$ are user-set parameters (default values are 0.3 and 0.5), it is assigned to a separate subgroup. Otherwise, it is assigned the nearest subgroup $i = \arg_{i \in \{1,2\}} \min \frac{|D(r,C_i,0,0)|}{|A(r,C_i,0,0)|}$. The purpose of the separate subgroup is to prevent the centroids from changing dramatically in each iteration.

3. We calculate the centroids for the first two subgroups and repeat step 2 until the three subgroups don't change.

After the group is divided into three subgroups, we repeat the above steps to continue dividing the subgroups. If the subgroup $G$ and its centroid $C$ meet the following conditions: $|G| \leq 3$ or $\frac{\sum_{r \in G} |D(r,C,0,0)|}{\sum_{r \in G} |A(r,C,0,0)|} \leq p_4$ and $\sum_{r \in G} |D(r,C,0,0)| \leq p_5$, it is no longer divided

into small subgroups. $p_4$ and $p_5$ are user-set parameters (the default values are 0.02 and 4).

After dividing the groups, we get a set of subgroups $SG$. we combine a pair of subgroups into a big one if the centroids are close to each other. First, we create an empty list $L$ and add the subgroup that contains the template read to the list as the first element $L_1$. Then, we combine other subgroup $g$ to $L_1$, if it meets the conditions: $D(g,L_1,p_6,p_7) < \min(p_8,A(g,L_1,p_6,p_7) \cdot p_9)$ and $D(g,L_1,p_7/2) < A(g,L_1,p_7/2) \cdot p_9$, where $p_6$, $p_7$, $p_8$ and $p_9$ are user-set parameters (the default values are 0.66, 6, 4 and 0.2). The remaining subgroups are put into another list $L'$ and sort the subgroups in ascending order of distance with $L_1$ ($|D(L_1,L'_i,p_6,p_7)|$). For the subgroup $i$ in the list $L'$, we combine it to the subgroup $j$ in list $L$, if it meets the following conditions: (1) $j = \arg\max|S(i,j,p_6,0)| | j \in L$; (2) $D(i,j,p_6,p_7) < \max(p_8,A(i,j,p_6,p_7) \cdot p_9)$; (3) $D(i,j,p_6,0) < A(i,j,p_6,0) \cdot p_9$. Otherwise, the subgroup $i$ is added to the end of the list $L$.

After combining the subgroups, we think the reads in the same subgroup in the list $L$ are from the same haplotype or the same copy of the repeat. The SNP alleles in the read can be verified by the centroid of its subgroup, which helps to find inconsistent overlaps more accurately. In addition, since the group $L_1$ contains the template read $t$, we use the centroid of the group $L_1$ to correct SNP alleles in the template read $t$. Here, the read is corrected when it is treated as a template read. After all reads are corrected, we run the verifying and correcting SNP alleles step again to obtain more robust results. We run the step twice by default.

**Finding inconsistent overlaps.** In the last round of verifying and correcting SNP alleles, we identify inconsistent overlaps. After combining steps, for each template read $t$, we obtain a subgroup list $L$ and the first subgroup $L_1$ of $L$ contains the template read $t$. We consider a query read $r$ and the template read $t$ to be inconsistent if the query read $r$ and its subgroup $L_r$ ($r \in L_r$) meet the conditions:

$$|D(L_r,L_1,p_6,p_7)| \geq \max(p_8,|A(L_r,L_1,p_6,p_7)| \cdot p_9); \quad (1)$$

$$\left|D(r,L_1,0,0)\bigcap A(L_r,L_1,p_6,p_7)\right| \geq \max(p_8,|A(r,L_1,0,0) \bigcap A(L_r,L_1,p_6,p_7)| \cdot p_9). \quad (2)$$

The parameters $p_6$, $p_7$, $p_8$ and $p_9$ are mentioned above. We record the location and direction of the reads in the haplotype-collapsed contigs. The overlaps between the inconsistent read pairs are identified as inconsistent overlaps if the directions and distances of the reads in the overlaps do not conflict with those in the haplotype-collapsed contigs (The difference between two distances should be less than 1000 by default.). The distance of the reads is defined as the distance of the 3' ends of the reads. We record the inconsistent overlap information and SNP alleles in reads for subsequent steps.

**Combining Nanopore raw reads to identify inconsistent overlaps.** For Nanopore datasets, we combine the corrected reads and raw reads to identify inconsistent overlaps. We first identify inconsistent overlaps using corrected reads with the strict threshold that the pair of reads should contain 8 different SNP alleles. Then, we use raw reads to identify inconsistent overlaps with the loose threshold that the pair of reads should contain 6 different SNP alleles. Using raw reads to identify inconsistent overlaps is similar to that using corrected reads. For Nanopore reads, we use Clair3[15] to call heterozygous SNPs for each contig. Since PECAT doesn't change the names of the reads during error correction, inconsistent overlaps between raw reads can be regarded as inconsistent overlaps between corrected reads. Similar to using corrected reads, we also check whether the directions and distances of the reads in the overlaps conflict with those in the haplotype-collapsed contigs. We record the location of corrected reads in raw

reads during error correction. Therefore, we can calculate the distance of corrected reads in the haplotype-collapsed contigs by linear mapping. The distance threshold is set to $\max(2500,0.05*D_c)$ by default, where $D_c$ is distance of corrected reads in the haplotype-collapsed contigs.

**Fast string-graph-based assembler**

According to the characteristics of k-mer-based alignment and string graphs, we propose a fast string-graph-based assembler to balance the quality and speed of assembling. First, we use minimap2 with parameters "-X -g3000 -w30 -k19 -m100 -r500" to find candidate overlaps between corrected reads. Minimap2[35] with those parameters invokes the k-mer-based alignment. To reduce overhangs of overlaps, we extend the alignment to the ends of the reads with the diff[36] algorithm and filter out overlaps still with long overhangs. Next, we remove the overlaps whose reads are contained in other reads or with low coverage. A directed string graph is constructed using the remaining overlaps. Myers' algorithm[33] is used to mark transitive edges as inactive ones. We implement local alignment using the edlib[48] algorithm to calculate the identity of each active edge, which is defined as the identity of its related overlap. Only a few edges need to calculate identities for most of the edges are marked as inactive ones. The edges whose identities are less than the threshold are marked as low quality and removed. The threshold is determined by the formula $((m_1 - 6*1.253*MAD_1) + 2*(m_2 - 6*1.4826*MAD_2))/3$, where $m_1$ and $m_2$ are the mean and the median of identity of all active edges in the string graph and $MAD_1$ and $MAD_2$ are the mean absolute deviation and the median absolute deviation of them. This step also breaks some paths connected by low-quality edges in the graphs. To repair those paths, we check dead-end nodes whose outdegree or indegree are equal to 0. We calculate the identities of their transitive edges and reactivate the edge with the longest alignment and the identity greater than the threshold. Considering that some breaks in paths are caused by reads being contained by other reads from the different haplotype or the different copy of the repeat, we extend the dead-end nodes with contained reads to repair the breaks using the similar method. In this way, an appropriate string graph is constructed. After performing other simplifying processes, such as removing the ambiguous edges (tips, bubbles, and spurious links) in the graph, we identify linear paths from the graph and generate contigs.

**Improvement of best overlap graph algorithm**

Best overlap graph algorithm[50] only retains the best out-edge and the best in-edge of each node according to the overlap length. After removing transitive edges using Myers' algorithm[33], PECAT performs the best overlap graph algorithm to further simplify the string graph. However, the original algorithm is not suitable for diploid assembly. The SNP alleles in reads are more important than the overlap length to measure which edge is the best one. Although most inconsistent overlaps are removed, there remain some undetected inconsistent overlaps and the corresponding reads contain different SNP alleles. We improve the algorithm and use the following steps to determine the best edges. First, for the in-edges or out-edges of each node, we sort the edges in descending order of the edge score $s_i = (n_i^+ \cdot w - n_i^-, l_i)$. $n_i^+$ and $n_i^-$ are the numbers of two related reads containing the same or different SNP alleles, which are obtained in finding inconsistent overlaps step. $w$ is a user-defined parameter. Its default value is set to 0.5. $l_i$ is the overlap length. The first edge $e_0$ is marked as a candidate best edge. The other edges meeting the following conditions also are marked as the candidate best edges: (1) Its related read is inconsistent with the reads related to the candidate best edges. (2) Its score is not too much less than the first edge $e_0$, which means $s_0[0] - s_i[0] < \begin{cases} \max(C,s_i[0] \cdot R_1), s_i[0] \geq 0 \\ \max(C, -s_i[0] \cdot R_2), s_i[0] < 0 \end{cases}$, where $C$, $R_1$ and $R_2$ are

user-defined parameters (the default values are 4, 2, and 0.66). Next, if the edge is marked as a candidate best edge twice (in-edge and out-edge), it is selected as the best edge. Then, if the node has no best in-edge or best out-edge, its first in-edge or first out-edge is selected as the best one. Finally, the edges selected as the best edges are retained and the other edges are removed from the string graph.

## Generating two sets of contigs

After simplifying the string graph, PECAT identifies three structures from the string graph, as shown in Supplementary Fig. 29a. First, we use FALCON-Unzip's heuristic algorithm[4] to identify bubble structures. Generally, the paths in the bubble structure are from different haplotypes. Then, we also identify alternate branches, which meet the following conditions: (1) There are only two branches. (2) The shorter branch is linear and does not contain other branches. (3) More than 30% of reads in the shorter branch are inconsistent with the reads in the other branch. We think the two branches that meet the conditions are from different haplotypes. We identify the paths in the string graph and do not break the paths if they encounter bubble structures or alternate branches. This trick can increase the continuity of primary contigs. The other paths in bubble structures or the alternate branches output as alternate contigs. We also check each pair of arbitrary independent contigs. If more than 30% of reads in the shorter contig are inconsistent with the reads in another contig, the shorter contig is labeled as an alternate contig. Another contig not labeled as an alternate contig is outputted as a primary contig.

To generate a dual assembly, PECAT connects the paths in two adjacent bubble structures. In the step of inconsistent overlap identification, we have called SNP alleles in each read. The information is used to determine the pair of paths in adjacent bubble structures that should be connected. As shown in Supplementary Fig. 29b, we build two read groups $I_1$ and $I_2$ for in-edges $in_1$ and $in_2$ respectively. If the reads only overlap with the reads in in-edges $in_1$ or $in_2$, they are assigned to $I_1$ or $I_2$, respectively. If the reads overlap with the reads in in-edges $in_1$ and $in_2$ at the same time, the reads are assigned to the group whose centroid the reads are closer to. The concepts of centroid and distance are defined in the previous section on verifying and correcting SNP alleles. In the same way, we get read groups $O_1$ or $O_2$ for out-edges $out_1$ or $out_2$. If the centroid of $I_1$ or $I_2$ does not have a common heterozygous site with the centroid of $O_1$ or $O_2$, we randomly connect the paths of bubble structures. Otherwise, we choose the paths to minimize the distance between their corresponding read group centroids.

## Polishing two sets of contigs

After generating contigs in primary/alternate format or dual assembly format, we use corrected reads or raw reads to polish them. We map the reads to the contigs using minimap2. In previous steps, we record which read pairs are inconsistent and which reads construct the contigs. If the read is inconsistent with the reads used for constructing the contig, the related alignments are removed. This trick helps to reduce haplotype switch errors. After removing other low-quality alignments, we run racon[51] with default parameters to polish the contigs.

For the assemblies from the Nanopore sequences, we use Medaka (https://github.com/nanoporetech/medaka) to further improve the assembly quality. Its steps are similar to those polishing with racon. The key step is to filter the inconsistent alignments between the reads and the contigs according to the information of inconsistent overlap.

## Evaluation

To evaluate the effectiveness of correction methods fairly, we extract the 40X longest reads from corrected reads and then evaluate them. We map the reads to the reference genome using minimap2 with parameters "-c --eqx". It generates CIGAR strings. We scan the CIGAR strings to calculate the accuracy of each corrected data. To evaluate

the accuracy of the sequences in the difficult-to-map regions and low-complexity regions in the HG002 reference genome, we map the HG002 reference genome to GRCh38. The regions are located in the HG002 reference genome according to GIAB v2.0 genome stratification BED files and the alignment between the HG002 reference genome and GRCh38. Then, we calculate the accuracy of the sequences in these regions separately. To evaluate the haplotype-specific k-mers completeness, we calculate the percentage of parent-specific k-mers in 40X longest reads. Considering that there are 40X datasets, we only count k-mers with equal to or more than 4 occurrences. To evaluate the haplotype-specific k-mers consistency, we calculate the metric as $\sum \max(k_p, k_m) / \sum (k_p + k_m)$, in which $k_p$ and $k_m$ are the number of paternal and maternal haplotype-specific k-mers in each read.

To evaluate the performance of selecting supporting reads and the performance of finding inconsistent overlaps, we use Illumina reads of parents to classify long reads with a trio-binning algorithm[16]. We adjust the threshold to reduce false positives. The read is classified as the paternal read if it meets the condition: $\frac{k_p}{K_p} > \frac{k_m + \max(10, k_m \cdot 0.1)}{K_m}$, where $k_p$ and $k_m$ are the numbers of paternal and maternal haplotype-specific k-mers in the read, and $K_p$ and $K_m$ are the numbers of all paternal and maternal haplotype-specific k-mers. The read is classified as a maternal read if it meets the condition: $\frac{k_m}{K_m} > \frac{k_p + \max(10, k_p \cdot 0.1)}{K_p}$. Otherwise, it is classified to the untagged reads. When one read of the pair of reads is a maternal read and the other read is a paternal read or vice versa, we think the pair of reads is inconsistent.

We use the k-mer-based assembly evaluation tool merqury[43] to evaluate the diploid assemblies generated by each pipeline and use BUSCO[42] to evaluate the gene completeness of assemblies. The hamming error rate of the assemblies is calculated from the output of merqury. The details of the parameters used in this study are described in Supplementary Notes 4, 5. We use the reference-based evaluation tool Pomoxis (https://github.com/nanoporetech/pomoxis) to evaluate the base quality of diploid assemblies (Supplementary Note 6). The genome fraction is evaluated by QUAST[52] and its parameters are described in Supplementary Note 7.

We obtain the small variants (SNP and INDEL) sets from HG002 assemblies against GRCh38 using dipcall[53] and compare them against the HG002 GIAB benchmark to evaluate their precisions, recalls, and F1 scores using hap.py[54]. We called structural variant (SV) sets from HG002 assemblies against GRCh37 using hapdiff (https://github.com/KolmogorovLab/hapdiff) and compare them against the curated set of SVs in the HG002 genome[55] to evaluate their precisions, recalls, and F1 scores using truvari[56] (Supplementary Note 8).

## Reporting summary

Further information on research design is available in the Nature Portfolio Reporting Summary linked to this article.

## Data availability

All described datasets are obtained from public websites, except for *A. thaliana* (Col-0 × C24) which is generated using our in-house sequencing. It is available from NGDC at PRJCA011723. *S. cerevisiae* (SK × Y12), *A. thaliana* (Col-0 × Cvi-0), *D. melanogaster* (ISO1 × A4), *B. taurus* (Angus × Brahman), and *B. taurus* (Bison×Simmental) are available from NCBI at PRJEB7245, PRJNA314706, PRJNA558397, PRJNA432857, and PRJNA677946. HG002 using Nanopore R9 sequencing is available at Human Pangenome Reference Consortium (HPRC) [https://s3-us-west-2.amazonaws.com/human-pangenomics/index.html?prefix=T2T/scratch/HG002/sequencing/ont/]. HG002 using Nanopore R10 sequencing is available at HPRC [https://s3-us-west-2.amazonaws.com/human-pangenomics/index.html?prefix=submissions/5b73fa0e-658a-4248-b2b8-cd16155bc157--UCSC_GIAB_R1041_nanopore/HG002_R1041_UL/Guppy6/]. HG002 using Nanopore R10 duplex sequencing is available at HPRC [https://s3-us-west-2.amazonaws.com/human-pangenomics/index.html?prefix=

submissions/0CB931D5-AE0C-4187-8BD8-B3A9C9BFDADE--UCSC_HG002_R1041_Duplex_Dorado/Dorado_v0.1.1/stereo_duplex]. HG002 using PacBio HiFi sequencing is available from NCBI at PRJNA586863. GIAB v2.0 genome stratification BED files are available at GIAB. HG002 GIAB benchmark is available at GIAB. The curated set of SVs in the HG002 is available at GIAB. NCTC9006 and NCTC9024 are available from ENA at PRJEB6403. The details of the datasets used in this study are reported in Supplementary Note 1 and Supplementary Table 17. All assemblies are available at Zenodo [https://doi.org/10.5281/zenodo.10457427].

## Code availability

All source codes for PECAT are available at GitHub [https://github.com/lemene/PECAT] and Zenodo [https://doi.org/10.5281/zenodo.10799833].

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

## Acknowledgements

This work was supported in part by the National Key Research and Development Program of China (No. 2021YFF1201200); the National Natural Science Foundation of China under Grants (Nos. 62350004, 62332020); the Project of Xiangjiang Laboratory (No. 23XJ01011) to Jianxin Wang. This work was also supported in part by the US National Institute of Food and Agriculture (NIFA; Grant Number 2017-70016-26051 and 2023-70029-41309) and the US National Science Foundation (NSF; Grant Number ABI-1759856, MRI-2018069, MTM2-2025541) to Feng Luo. This work was also supported in part by Guangdong Basic and Applied Basic Research Foundation (No. 2020B1515020057) to Chuanle Xiao. We are grateful for resources from the High-Performance Computing Center of Central South University.

## Author contributions

J.X.W., F.L., and C.L.X. conceived and designed this project. F.N. and J.X.W. conceived, designed, and implemented the consensus and assembly algorithm. F.N. and P.N. integrated all the programs into the PECAT pipeline and provided documentation. F.N., P.N., N.H., and J.Z. analyzed the performance of algorithms developed in this study. C.L.X. and Z.Y.W. provided the dataset *A. thaliana* (Col-0 × C24). J.X.W., F.L., and F.N. performed the theoretical analysis of the algorithms developed in this study. J.X.W., F.L., and F.N. wrote the manuscript. Z.Y.W. wrote about cultivating *A. thaliana* (Col-0 × C24) and extracting its DNA. All authors have read and approved the final version of this manuscript.

## Competing interests

The authors declare no competing interests.
