## [Peer Review File · Nature Communications]

de novo diploid genome assembly using long noisy readsREVIEWERS' COMMENTS

Reviewer #1 (Remarks to the Author):

Thank you for the authors for the revisions. I understand that this has been a long and tedious review process. I am happy with all the results presented in the manuscript.

The only issue I am raising is removal of the two tables in response to the comment on "read correction" of highly accurate reads.

I would strongly recommend the authors to add back in the results for "correcting highly accurate reads" and discuss the results as a limitation of this method like they did in the response. I was not looking for a solution to the allele-specific error correction with PECAT, just stating that it remains a problem in this field and we need future work is a great way to showcase where error correction is most needed and where it can cause harm.

I thank the authors again for the careful reviews. I am satisfied with the manuscript once this minor concern is addressed.

Summary

We appreciate the valuable comments and suggestions from the editor and reviewers. Based on the suggestions and comments from editor and reviewers, we revised our manuscript. We addressed those comments and suggestions carefully. The significant changes in the revised manuscript were highlighted by red color.

Answers to Reviewer #1

Reviewer #1 (Remarks to the Author):

Thank you for the authors for the revisions. I understand that this has been a long and tedious review process. I am happy with all the results presented in the manuscript.

Authors' Response. We are very pleased that our work has been recognized. We appreciate your helpful suggestions.

The only issue I am raising is removal of the two tables in response to the comment on "read correction" of highly accurate reads.

Authors' Response. The metrics of PECAT correction method on highly accurate reads, including Nanopore duplex reads and HiFi reads, are shown in Supplementary Table 11.

I would strongly recommend the authors to add back in the results for "correcting highly accurate reads" and discuss the results as a limitation of this method like they did in the response. I was not looking for a solution to the allele-specific error correction with PECAT, just stating that it remains a problem in this field and we need future work is a great way to showcase where error correction is most needed and where it can cause harm.

Authors' Response. We added back the results of error correction by PECAT on Nanopore duplex reads and HiFi reads to the "Performance on highly accurate long reads" Section. We have discussed the shortcomings of PECAT's error correction method in the Discussion section.

I thank the authors again for the careful reviews. I am satisfied with the manuscript once this minor concern is addressed.

Authors' Response. Thanks again for your support.